# VOC Gas Sensors Based on Zinc Stannate Nanoparticles Decorated with Silver

**DOI:** 10.3390/nano14241993

**Published:** 2024-12-12

**Authors:** Svetlana S. Nalimova, Zamir V. Shomakhov, Dmitry A. Kozodaev, Arina A. Rybina, Sergey S. Buzovkin, Cong D. Bui, Ivan A. Novikov, Vyacheslav A. Moshnikov

**Affiliations:** NT-MDT BV, 7335 Apeldoorn, The Netherlands; shozamir@yandex.ru (Z.V.S.); kozodaev@ntmdt.nl (D.A.K.); arinasvg02@gmail.com (A.A.R.); sergey.bu2015@gmail.com (S.S.B.); congdoan6997@gmail.com (C.D.B.); i.novikov@ntmdt.nl (I.A.N.); vamoshnikov@mail.ru (V.A.M.)

**Keywords:** gas sensor, nanomaterial, zinc stannate, noble metal, sensitization

## Abstract

Today, air pollution is a global environmental problem. A huge amount of explosive and combustible gas emissions that negatively affect nature and human health. Gas sensors are one of the ways to prevent this impact. Several types of sensors have been developed, but the main problem with them is the high operating temperature. This leads to a decrease in reproducibility and stability over time. The aim of the work is to synthesize zinc stannate nanoparticles, study their phase composition, and modify the structure with silver nanoparticles to improve gas-sensing characteristics. This paper presents the synthesis of zinc stannate nanoparticles by the coprecipitation method and annealing at varying temperatures. A method of decorating zinc stannate with silver nanoparticles was proposed. Using XRD, it was found that a change in the annealing temperature leads to a change in the initial phase composition of the sample. Decoration with silver nanoparticles allows for increasing the sensor response of zinc stannate layers to isopropyl alcohol by 20 times. The corresponding increase in response to ethanol is 16 times.

## 1. Introduction

Today, gas sensors play a crucial role in various industries, including medicine and environmental protection. A typical chemiresistive gas sensor consists of a substrate, metal electrodes, sensing material and a heating element [1]. In order to detect a signal, a resistance measurement system is required. The main characteristics of semiconductor gas sensors include selectivity to the target gas, sensitivity, response and recovery times, as well as stability over time and temperature [2].

The most common semiconductors used in gas sensors are zinc oxide (ZnO) and tin dioxide (SnO_2_). In previous studies [2,3], it has been found that the ternary oxide structure of zinc stannate is more effective. Zinc stannate, with its band gap of 3.6 eV and n-type conductivity, has high electron mobility and electrical conductivity, as well as a low coefficient of thermal expansion. This material has a wide range of applications [4], including gas sensors, transparent electrodes for solar cells and lithium–ion batteries.

Zinc stannate can exist in two different crystal structures [5], as shown in Figure 1. One structure is perovskite (orthorhombic symmetry) [6], ZnSnO_3_, and the other is inverse spinel (cubic symmetry), Zn_2_SnO_4_ [7]. At low synthesis temperatures, the perovskite structure is formed. However, this structure is unstable, and under external influences, such as a temperature increase to 500 °C, it transforms into an amorphous phase, leading to the formation of oxygen vacancies that increase the sensitivity of gas sensors [8,9,10]. As the temperature continues to rise, recrystallization takes place and stable zinc orthostannate (Zn_2_SnO_4_) with inclusions of SnO_2_ forms. Typically, synthesized powders contain a mixture of both perovskite and inverse spinel phases in varying proportions [11].

Currently, there are several methods for producing zinc stannate, including hydrothermal synthesis [12,13], ion exchange [14], sol–gel technique [15], and thermal precipitation [16,17]. One of these methods, coprecipitation [18], has been chosen for this study.

The main goal of further development is to reduce operating temperatures while maintaining sensor sensitivity [19]. This is important because high temperatures can lead to several problems, such as shorter sensor lifetime due to material degradation and deformation caused by differences in thermal expansion coefficients between the sensor layer and substrate. Additionally, using heating elements can complicate the device, and VOC decomposition at high temperatures can affect the stability and accuracy of the sensor.

To address these issues, noble metals can be used for decoration [20], which can help to reduce operating temperatures. The decoration with noble metal nanoparticles increases the specific surface area of the sensor, providing an additional active adsorption site [21]. This, in turn, enhances the sensor’s sensitivity and selectivity. The use of noble metals, such as palladium (Pd), in the decoration of a semiconductor allows the sensor to selectively detect hydrogen (H_2_) molecules [22,23]. This is because Pd has a strong affinity for hydrogen and can selectively adsorb and dissociate hydrogen molecules from other gases. Upon interaction with hydrogen, Pd undergoes a specific reaction that allows for its detection:2Pd + xH_2_ → 2PdH_x_.(1)

PdH has a lower work function than Pd. This means that the Schottky barrier between the two materials is lower. As a result, more electrons can be injected into the semiconductor. This leads to a larger change in resistance.

The aim of this work is to synthesize zinc stannate layers at different annealing temperatures, modify them with silver nanoparticles, and study the gas-sensing properties of the resulting structures.

## 2. Materials and Methods

### 2.1. Sample Fabrication

Coprecipitation in an aqueous solution was selected as a technique for producing zinc stannate. This method allows for the production of a nanopowder with a high degree of phase uniformity. Zinc acetate dihydrate (Zn(CH_3_COO)_2_·2H_2_O) and sodium stannate trihydrate (Na_2_SnO_3_·3H_2_O) were chosen as precursors. Using an ultrasonic bath, 0.02 M Zn(CH_3_COO)_2_·2H_2_O and 0.02 M Na_2_SnO_3_·3H_2_O were dissolved in 100 mL of distilled water and stirred at 50 °C for 20 min on a magnetic stirrer. Furthermore, the resulting solutions were mixed at the same temperature for 2 h. A white precipitate was separated using a centrifuge. Then it was washed three times with distilled water and dried at 80 °C in an air atmosphere. The resulting powder was annealed at 300 °C, 500 °C and 700 °C for two hours in a muffle furnace [18]. The corresponding samples were marked as ZTO-300, ZTO-500, ZTO-700.

### 2.2. Preparation of Ag Nanoparticles and Decoration of Zinc Stannate

To produce silver nanoparticles, ascorbic acid, silver nitrate and sodium citrate were used as starting materials [24]. The synthesis process involved the following steps. Aqueous solutions of silver nitrate (2 M), ascorbic acid (1.5 M) and sodium citrate (0.05 M) were prepared. A total of 10 mL of acetone and 10 mL of ethanol were mixed using a magnetic stirrer. Silver nitrate (0.5 mL), ascorbic acid (0.25 mL) and sodium citrate (0.5 mL) solutions were added to the solution of acetone and ethanol and mixed for 15 min. The precipitate was washed three times with distilled water and dried in the air at 70 °C.

A total of 50 mg of silver nanoparticles were dispersed in 50 mL of distilled water. Then, 1 g of ZTO-500 was added to the resulting suspension, which was mixed for 2 h. The precipitate was collected using a centrifuge and dried in the air at 70 °C. The corresponding sample was marked as ZTO_Ag-500.

### 2.3. Characterization of Sample

The crystal structure of the samples was studied by X-ray difraction (XRD) with Cu-Kα radiation (D2 Phaser, Bruker AXS, Karlsruhe, Germany). Scanning electron microscopy (SEM) was chosen to study the morphology of the samples, elemental composition was analyzed by energy dispersive spectroscopy (Tescan VEGA3LMH, Tescan, Brno, Czech Republic). X-ray photoelectron spectroscopy (XPS) was used to study the elemental composition of the samples (K-Alpha, Thermo Scientific, Waltham, MA, USA) using a monochromatic Al Kα X-ray source (1486.6 eV), and the residual pressure in the analytical chamber was approximately 4 × 10^−9^ mbar. Atomic force microscopy (AFM) was used to estimate the average size of silver nanoparticles (NTEGRA Aura, NT-MDT, Apeldoorn, The Netherlands).

### 2.4. Sensor Fabrication and Measurement

The resulting powders were deposited onto ceramic sensor platforms with gold electrodes in an interdigitated configuration (Appendix A). The distance between the electrodes was 200 μm, and their width was 200 μm. The substrates were first cleaned with acetone, isopropyl alcohol, and distilled water to remove any impurities. The powders were dispersed in 10 mL of water and deposited on the substrates by spin-coating. The coated substrates were then annealed at 300 °C (ZTO-300), 500 °C (ZTO-500) and (ZTO-500_Ag) and 700 °C (ZTO-700) for 15 min in an air atmosphere.

Gas-sensing properties were measured using a special stand (Figure 2). All measurements were performed at a voltage of 5 V.

The current through the sample was measured using a Keithley 6485 picoammeter (Keithley, Cleveland, OH, USA). An electromechanical valve, controlled by a computer, allowed switching between two flow rates. When the valve is off, dry air is supplied to the measurement chamber, and when it is on, a mixture of dry air and the target gas is introduced. The flow rates were monitored using rotameters. The vapor concentration of the target gas was determined by the saturation vapor pressure at room temperature:(2)$$C=\frac{P_{gas}F_{gas}}{P_{atm}\left(F_{gas}+F_{air}\right)},$$
where P_gas_ is the saturated vapor pressure of the bubbled liquid, F_gas_ is the air flow rate through the bubbler, P_atm_ is atmospheric pressure, and F_air_ is the diluent air flow rate. Measurements were taken when the samples were exposed to acetone, ethanol, and isopropanol.

The response was calculated as the ratio of the sensor’s resistance in air to its resistance in the presence of a reducing gas at a given concentration:(3)$$S=\frac{R_{air}}{R_{gas}},$$
where R_air_ is the resistance of the sensor in the air atmosphere, R_gas_ is the resistance of the sensor in the atmosphere of the target gas.

Response and recovery times refer to the time it takes for the sensor’s resistance to change by 90% after target gas and air have been supplied, respectively.

## 3. Results

Figure 3 shows the results of X-ray diffraction of non-annealed powder, ZTO-300, ZTO-500 and ZTO-700. Then it can be seen that the non-annealed powder is zinc hydroxostannate ZnSn(OH)_6_. All of the peaks correspond to the standard cubic phase of ZnSn(OH)_6_ (JCPDS file No. 73-2384, space group: Pn-3 (201)) [25]. No diffraction peaks due to impurities or other phases were observed. The sharp and narrow diffraction peaks suggest the high crystallinity of the zinc hydroxostannate.

During the annealing of the precursor powder at 300 °C and 500 °C, the crystallinity has been disrupted. Analysis of XRD patterns of powders annealed at 300 °C and 500 °C showed the formation of low crystalline structures at these temperatures. The wide peaks at 34° and 60°, corresponding to the crystallographic planes (110) and (211) of face-centered perovskite ZnSnO_3_ (JCPDS 11-0274) [26], indicate the presence of zinc stannate nanoparticles with a small size. The XRD pattern of the ZTO-700 sample reveals a well-defined crystal structure. It was found that the cubic inverse spinel Zn_2_SnO_4_ phase (JCPDS, No. 73-1725) and tetragonal SnO_2_ phase (JCPDS, No. 77-0451) were formed at 700 °C [27]. In this way, a phase transition can be observed.

The results of the studies on the morphology of zinc stannate powders using SEM are shown in Figure 4. As can be seen, the particles have a cubic shape. The average particle size is the same after annealing both at 300 °C and 500 °C.

The uniform distribution of Sn, Zn and O in the synthesized nanoparticles after annealing at 500 °C is shown in Figure 5.

The sizes of the synthesized silver nanoparticles were studied using AFM, as shown in Figure 6. The average particle size was found to be 70 nm.

The EDS spectrum (Figure 7a) confirms the presence of Ag in the modified sample. At the same time, the percentages of elements determined by the spectrum are 51.5 wt.% (Sn), 24.2 wt.% (Zn), 15.4 wt.% (O), and 8.9 wt.% (Ag). The EDX scan (Figure 7b) shows the uniform distribution of silver particles over the surface of the sensor layer.

The surface elemental composition and electronic structure of the synthesized ZTO-300, ZTO-500 and ZTO-700 powders were analyzed by XPS. Distinct peaks can be seen in the XPS survey spectrum (Figure 8). These peaks correspond to the binding energies of zinc, tin, oxygen and carbon [28].

In the O1s spectra (Figure 9), two peaks can be distinguished, corresponding to energies of 530.3 eV and 531.5 eV. The more intense peak of 530.3 eV corresponds to oxygen in the crystal lattice (O_lat_) [29], and the second peak of 531.5 eV refers to oxygen vacancies on the surface (O_vac_) [30]. The percentage ratio between different forms of oxygen on the surface of samples was analyzed. A decrease in the proportion of oxygen, corresponding to oxygen vacancies, was found as the annealing temperature increased (47% O_vac_ for ZTO-300, 34% O_vac_ for ZTO-500, 29% O_vac_ for ZTO-700).

The successful decoration of Ag nanoparticles on the surface of zinc stannate nanoparticles was confirmed by XPS. The study of the Ag 3d X-ray photoelectron spectrum (Figure 10) revealed the presence of two peaks with binding energies at 374.4 eV and 368.4 eV, corresponding to the 3d3/2 and 3d5/2 states of metallic silver, respectively [31,32].

The gas-sensing properties of the samples heated to 250 °C were studied and are presented in Table 1. The results showed that ZTO-300 had the highest response to isopropyl alcohol (6.26), while ZTO-500 had the highest response to acetone (7.1). ZTO-700, however, did not exhibit distinct selectivity.

Based on the results obtained, it can be concluded that the samples consisted of ZnSnO_3_ show higher responses compared to multiphase sample of Zn_2_SnO_4_ and SnO_2_. Therefore, it is likely that the Zn_2_SnO_4_ and SnO_2_ phases formed at 700 °C do not form a heterojunction. Another possible explanation for this is the nonuniform distribution of these phases in the powder.

A semiconductor acts as the active layer in a gas sensor. When it is exposed to different gases, the resistance of the sensor changes. This change depends on the type of semiconductor and the nature of the gas [33].

For instance, zinc stannate is an n-type semiconductor. If reducing gases such as CO, NH_3_, and volatile organic compounds are present, resistance decreases. Conversely, if oxidizing gases like O_2_ and NO_2_ are present, resistance increases. P-type semiconductors have the opposite reaction.

This type of resistance change can be explained by the formation of a “core–shell” structure as a result of oxygen adsorption. The electrical resistance of the sensor is due to the formation of a contact between the shells of grains (Figure 11).

The gas detection process consists of several stages, including adsorption, oxidation, and desorption. When a sensor is exposed to air, oxygen molecules are adsorbed onto its surface. Oxygen has a higher electron affinity than the work function of the semiconductor, so electrons move from the conduction band to the oxygen. This causes the surface to become negatively charged. The form of adsorbed oxygen depends on the number of free electrons and temperature [34]:(4)$$O_{2}\left(gas\right)\to O_{2}\left(ads\right),$$
(5)$$O_{2}\left(ads\right)+e^{-}\to {O_{2}}^{-}\left(ads\right)\left(T<150\;{}^{\circ}C\right),$$
(6)$$O_{2}\left(ads\right)+2e^{-}\to {2O}^{-}\left(ads\right)\left(150\;{}^{\circ}C\;<\;T<\;400\;{}^{\circ}C\right),$$
(7)$$O_{2}\left(ads\right)+{4e}^{-}\to {2O}^{2-}\left(ads\right)\left(T>400\;{}^{\circ}C\right).$$

Then, in the near-surface layer of the semiconductor, the concentration of free charge carriers decreases and a depleted layer is formed. This layer has increased resistance, creating a shell. As a result, free charge carriers are redistributed in the semiconductor, leading to bending of energy bands and the formation of a potential barrier. This causes the resistance of the semiconductor to increase.

The thickness of depleted layer and corresponding bending of the energy bands depends on the Debye length [35,36]:(8)$$L_{D}=\sqrt{\frac{{2\varepsilon \varepsilon }_{0}U_{s}}{qN}},$$
where ε is the relative permittivity of the semiconductor, ε_0_ is the vacuum permittivity, U_s_ is the height of the potential barrier, q is the elementary charge, N is the concentration of ionized donors.

When reducing gases are supplied, their molecules are adsorbed onto the surface of the semiconductor. Then redox reactions occur between the adsorbed gas particles and the adsorbed oxygen:(9)$$\;VOCs\;\left(gas\right)\to VOCs\left(ads\right),$$
(10)$$VOCs\left(ads\right)+O^{-}\backslash {O_{2}}^{-}\to CO_{2}+H_{2}O\left(gas\right)+e^{-}.$$

This reduces the resistance of the material, as well as the height of the potential barrier and thickness of the depleted layer.

A possible explanation for the observed dependencies may be related to the different mechanisms of interaction between metal oxides, acetone, and alcohols. According to XRD results, ZTO-300 and ZTO-500 samples have a similar crystal structure. However, there may be differences in the ratio of zinc, tin, and oxygen ions on the surfaces of these samples. It is known that acetone molecules can be adsorbed on metal cations [37,38], while alcohol adsorption can occur on both metal cation sites and hydroxyl groups [39,40]. The high concentration of metal cation sites on ZTO-500 surface, which serve as adsorption sites for acetone molecules, could explain why the response to acetone is higher than the response to alcohol. Conversely, the higher response of ZTO-300 sample to isopropyl alcohol may be due to a higher content of oxygen vacancies on its surface, which can adsorb hydroxyl groups from ambient air. The lower responses of the ZTO-700 sample are associated with the formation of two phases, Zn_2_SnO_4_ and SnO_2_.

The ZTO-500 sample, which showed a high gas-sensing response, was chosen for decoration with silver nanoparticles. After decoration, it was also tested for gas sensitivity. The results are presented in Table 2. The typical response curve of ZTO-500_Ag to isopropyl alcohol is shown in Figure 12. The effect of water vapor on the response of ZTO-500_Ag when detecting isopropyl alcohol at a concentration of 1000 ppm showed that at a relative humidity of 10%, the response was reduced by 17%. The response of the ZTO-500_Ag to isopropyl alcohol at 250 °C depends on the concentration of the target gas, as shown in Figure 13. There is a linear increase in response as the concentration increases. The stability of the baseline resistance was studied at a temperature of 250 °C for 12 h. The results are presented in Appendix A. During this time, the maximum deviation of the resistance from its initial value was 4%.

When comparing the results of gas sensitivity tests, it is can be seen that the responses to isopropyl alcohol and ethanol have increased by more than an order of magnitude. This confirms the theory of increasing the sensitivity of gas sensors through decoration with noble metals. Silver nanoparticles have acted as catalysts for chemical adsorption, as expected.

In addition to the fact that silver particles may increase sensitivity, they may also lower operating temperatures. To test this, a gas sensitivity to isopropyl alcohol was studied at a lower temperature of 150 °C (Figure 14). Under these conditions, the response was six, with a response time of 210 s and a recovery time of 450 s.

Thus, when the temperature was reduced by 100 °C, ZTO-500_Ag exhibited a response similar in magnitude to that of ZTO-500 before modification with silver nanoparticles at 250 °C.

The improvement of gas-sensitive properties is associated with two main mechanisms, chemical and electronic sensitization [41,42]. Electronic sensitization occurs when charge carriers (electrons) transfer at the metal–semiconductor interface due to the difference in work functions between the electrons in the metal and semiconductor. When electrons diffuse, they redistribute charge carriers and disrupt the electrical neutrality at the interface. With this diffusion of electrons, the charge carriers are redistributed and the electrical neutrality of the regions at the interface is disrupted.

Consequently, a contact potential difference arises, which causes the bending of the bands and the formation of a potential barrier. The height of the barrier formed depends on the contact potential difference, which is determined by the difference between the work functions of the metal and semiconductor. Figure 15 illustrates the band diagram for the metal–semiconductor junction. The work function of a noble metal is higher than that of a semiconductor, so electrons will transfer from the conduction band of a semiconductor to the metal. As a result, a Schottky barrier forms, which increases the electron-depleted layer. This can prevent the recombination of electrons and holes, and cause a higher change in resistance when the sensor is exposed to reducing gases. As a result, the response of the sensor increases.

In chemical sensitization (Figure 16) noble metals act as catalysts for chemical adsorption, reducing the adsorption energy of gases and contributing to the dissociation of oxygen molecules into more reactive oxygen atoms. These formed ions then transport from the noble metal to the surface of the semiconductor, where they react with the target gas. When the target gas is present, a larger number of oxygen ions enter redox reactions, releasing a greater number of electrons. Consequently, the response of the sensor increases.

The observed decrease in the response of ZTO-500 to acetone after modification with silver nanoparticles can be caused by the influence of chemical sensitization. As a result, the concentration of ionosorbed oxygen ions, occupying free metal cation sites on the surface, increases. The content of adsorption sites for acetone molecules decreases; therefore, the response to acetone also decreases.

At the same time, as discussed earlier, the adsorption of alcohol molecules can occur with the participation of surface hydroxyl groups, so the concentration of adsorbed alcohol molecules remains high. An increase in the concentration of ionosorbed oxygen leads to an increase in the number of electrons formed as a result of the reaction (10) and an increase in the responses to isopropyl alcohol and ethanol.

A comparison of the gas-sensing characteristics achieved in this study with the results of other researchers on the development of silver-modified sensors based on metal oxide materials is presented in Table 3.

The table shows silver modification is an effective approach for developing highly efficient sensors. The findings from this study demonstrate the ability to achieve response and recovery at temperatures significantly lower than those reported in most previous studies.

## 4. Conclusions

Three structures of zinc stannate were obtained using the coprecipitation method in an aqueous solution. When studying their gas-sensitive properties, it was found that ZTO-300 was selective to isopropyl alcohol vapors, ZTO-500 to acetone vapors and ZTO-700 showed no selectivity. XRD analysis found that ZT0-300 and ZTO-500 had an amorphous-like structure, while ZTO-700 had a crystalline structure of Zn_2_SnO_4_ and SnO_2_. SEM analysis revealed that all samples had a cubic structure, and the average particle size was 180 nm, regardless of annealing temperature. A sample decorated with silver nanoparticles showed a 20-fold improvement in gas sensitivity to isopropyl alcohol, and reducing operating temperatures to 150 °C was possible by decorating with silver nanoparticles. Using AFM, it was found the average size of the silver nanoparticles was 70 nm.

## Figures and Tables

**Figure 1 nanomaterials-14-01993-f001:**
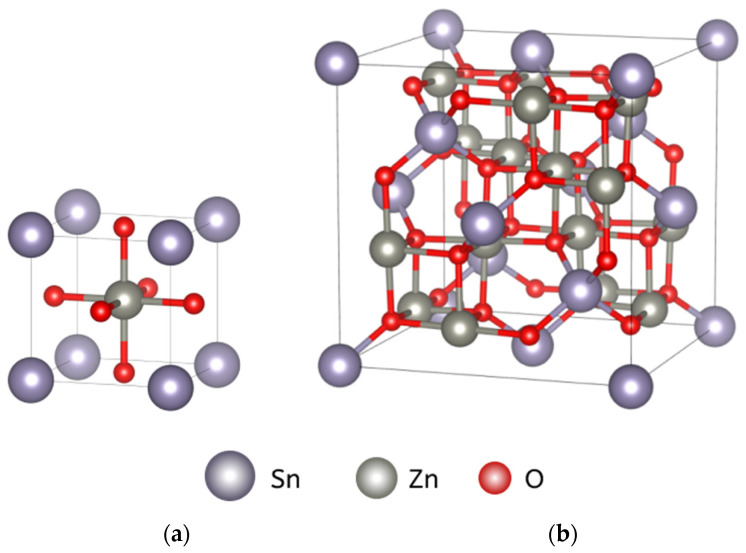
Crystal structures of ZnSnO_3_ (**a**) and Zn_2_SnO_4_ (**b**).

**Figure 2 nanomaterials-14-01993-f002:**
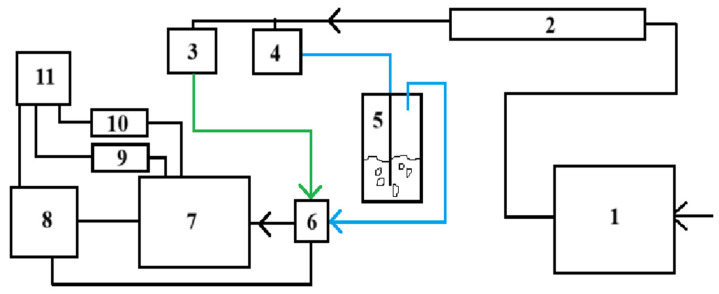
The scheme of the measuring stand: 1—compressor, 2—dehumidifier, 3, 4—rotameters, 5—bubbler, 6—electromechanical valve, 7—measuring cell, 8—controller, 9, 10—picoammeter and voltage source, 11—personal computer.

**Figure 3 nanomaterials-14-01993-f003:**
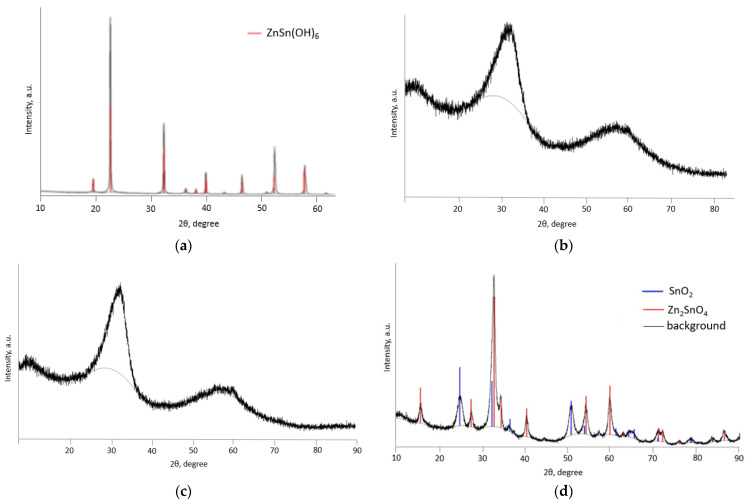
XRD patterns of: (**a**) non-annealed powder; (**b**) ZTO-300; (**c**) ZTO-500; (**d**) ZTO-700.

**Figure 4 nanomaterials-14-01993-f004:**
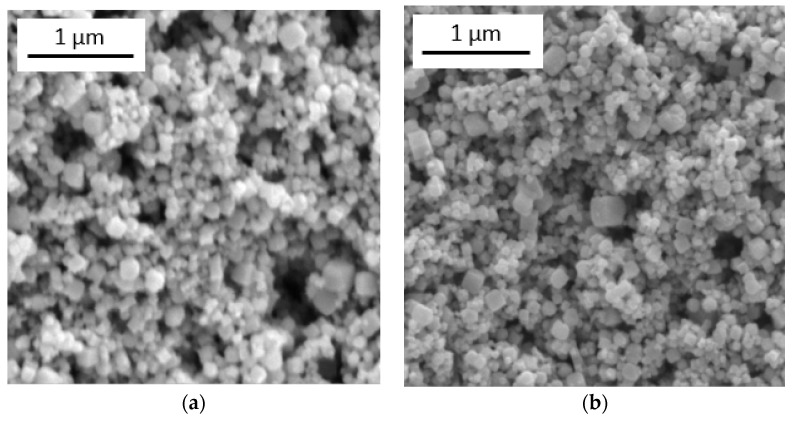
SEM images of: (**a**) ZTO-300; (**b**) ZTO-500.

**Figure 5 nanomaterials-14-01993-f005:**
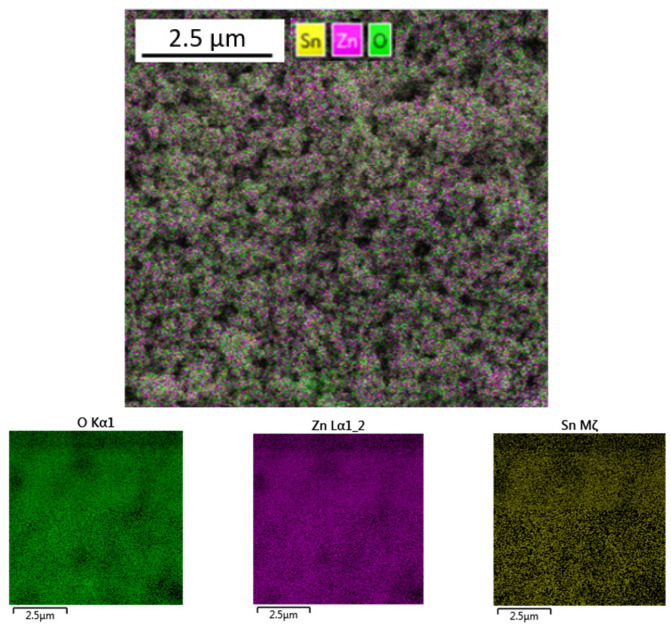
Elemental mapping for ZTO-500.

**Figure 6 nanomaterials-14-01993-f006:**
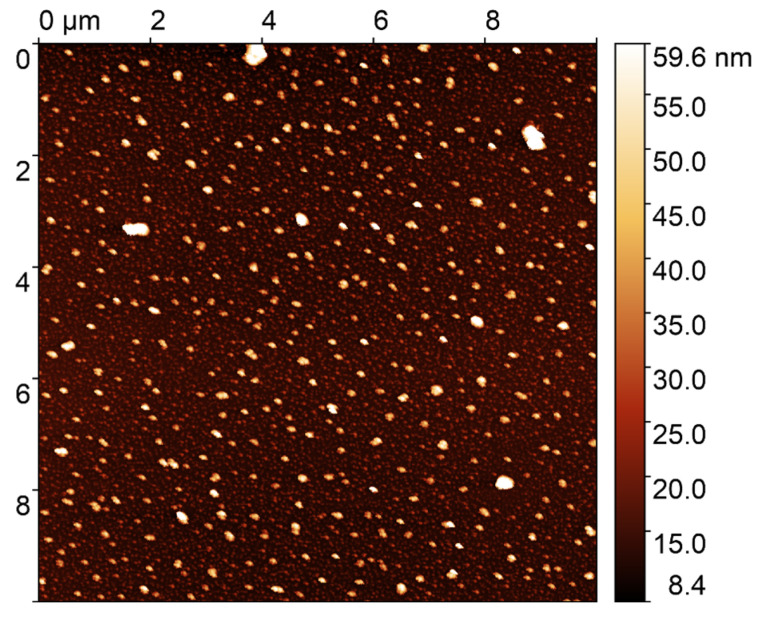
AFM image of silver nanoparticles.

**Figure 7 nanomaterials-14-01993-f007:**
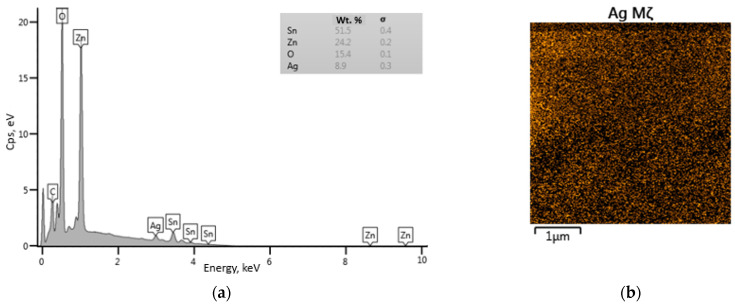
(**a**) EDS spectrum of ZTO-500_Ag; (**b**) Elemental mapping of Ag in ZTO-500_Ag.

**Figure 8 nanomaterials-14-01993-f008:**
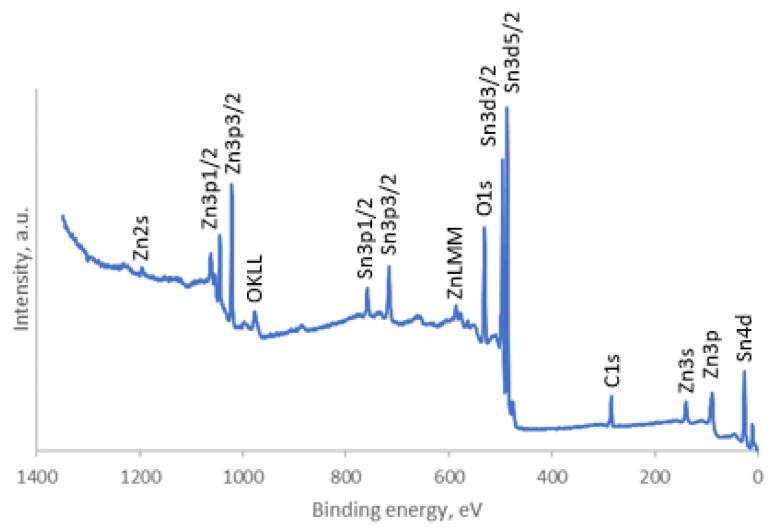
Survey X-ray photoelectron spectrum of ZTO-500.

**Figure 9 nanomaterials-14-01993-f009:**
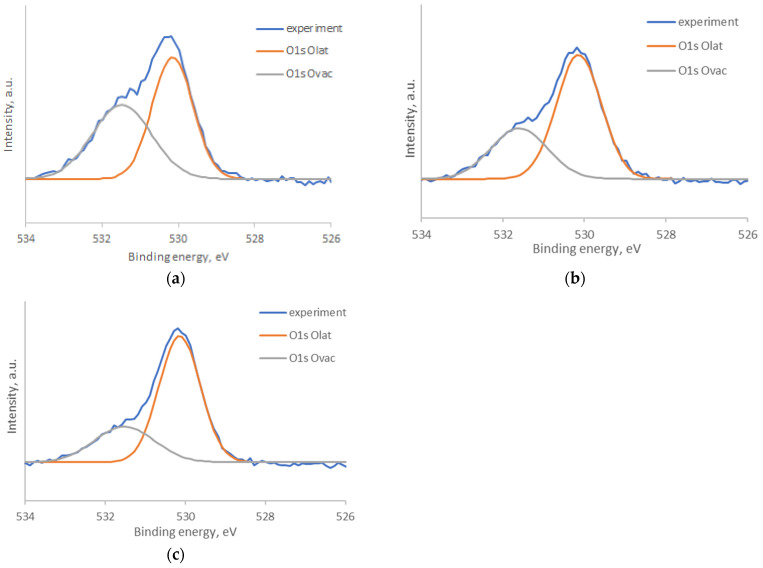
X-ray photoelectron spectra of O1s level: (**a**) ZTO-300; (**b**) ZTO-500; (**c**) ZTO-700.

**Figure 10 nanomaterials-14-01993-f010:**
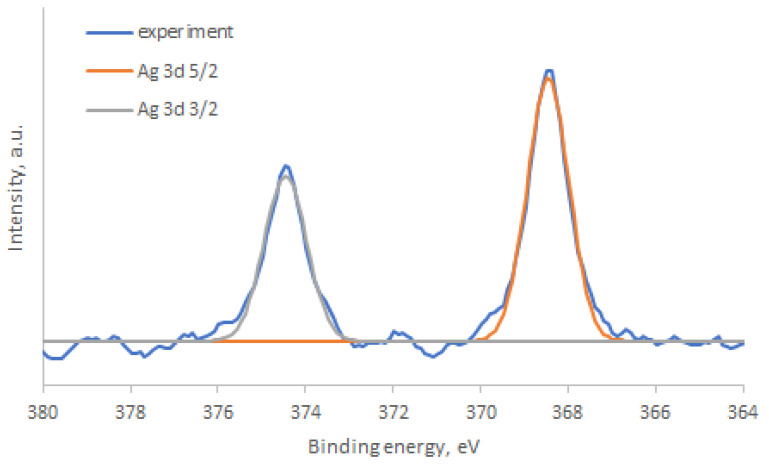
X-ray photoelectron spectra of Ag3d level in ZTO-500_Ag.

**Figure 11 nanomaterials-14-01993-f011:**
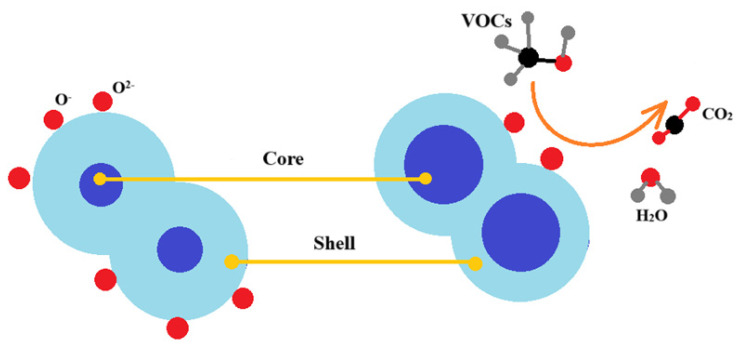
The mechanism of gas sensitivity in zinc stannate.

**Figure 12 nanomaterials-14-01993-f012:**
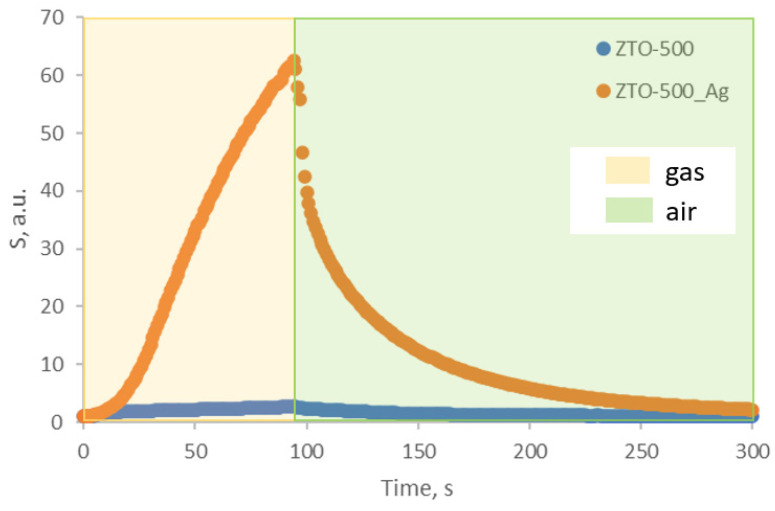
ZTO-500_Ag and ZTO-500 response curves to isopropyl alcohol with a concentration of 1000 ppm at 250 °C.

**Figure 13 nanomaterials-14-01993-f013:**
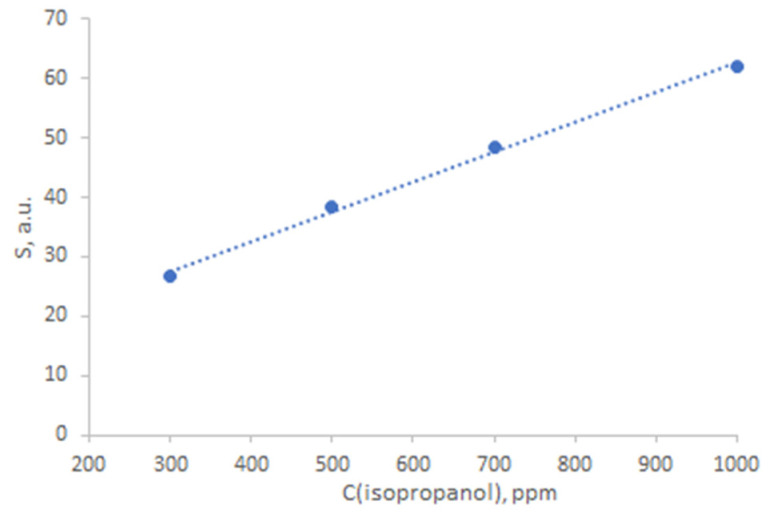
Concentration dependence of ZTO-500_Ag response to isopropyl alcohol at 250 °C.

**Figure 14 nanomaterials-14-01993-f014:**
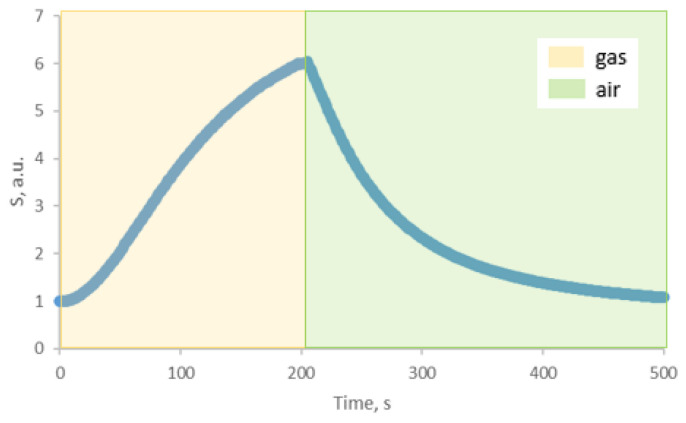
ZTO-500_Ag response curve to isopropyl alcohol with a concentration of 1000 ppm at 150 °C.

**Figure 15 nanomaterials-14-01993-f015:**
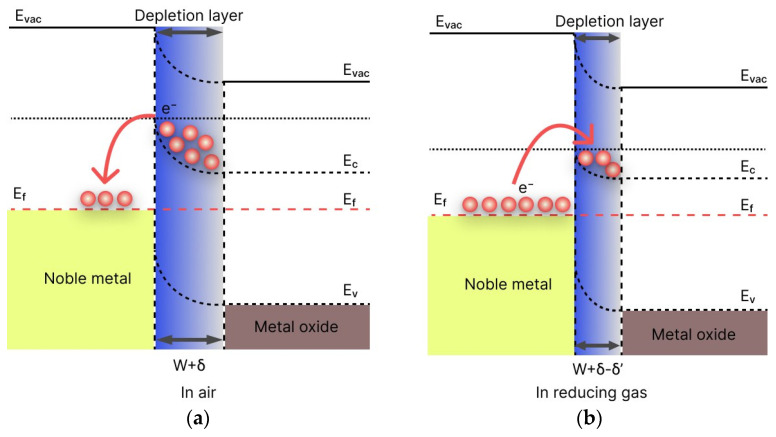
Band diagram of the metal–semiconductor interface: (**a**) in air; (**b**) in the presence of reducing gas [20].

**Figure 16 nanomaterials-14-01993-f016:**
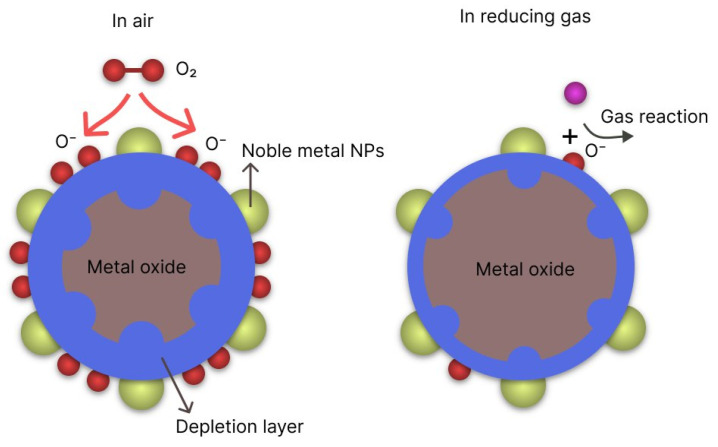
Chemical sensitization mechanism [20].

**Table 1 nanomaterials-14-01993-t001:** Experimental data on the gas-sensing properties of ZTO for different VOCs at a concentration of 1000 ppm.

Structure	Gas	Response	Response/Recovery Time, s
ZTO-300	isopropyl alcohol	6.26	143/313
	acetone	1.56	143/88
	ethanol	1.59	61/110
ZTO-500	isopropyl alcohol	3	108/254
	acetone	7.1	18/282
	ethanol	2.2	34/235
ZTO-700	isopropyl alcohol	1.48	62/60
	acetone	2.37	106/159
	ethanol	1.75	179/120

**Table 2 nanomaterials-14-01993-t002:** Comparison of the results from measuring the gas sensitivity of ZTO-500 before and after decoration with silver nanoparticles.

	Isopropyl Alcohol (1000 ppm)		Acetone (1000 ppm)		Ethanol (1000 ppm)	
	ZTO-500	ZTO_Ag-500	ZTO-500	ZTO_Ag-500	ZTO-500	ZTO_Ag-500
Response	3	62	7.1	4.22	2.2	35.6
Response time, s	177	58	60	220	34	73
Recovery time, s	254	457	282	238	235	442

**Table 3 nanomaterials-14-01993-t003:** Gas-sensing performances of the developed sensor and reported in the literature.

Sensor Material	Target Gas (Concentration)	Response, R_a_/R_g_	Temperature, °C	Reference
Ag-ZnO	H_2_, 300 ppm	5.79	250	[43]
Ag-ZnO	Ethanol, 100 ppm	148	320	[44]
Ag-ZnO/In_2_O_3_	HCHO, 100 ppm	186	260	[45]
Ag-In_2_O_3_	Isopropanol, 5 ppm	5.2	300	[46]
Ag-ZnO	NO_2_, 10 ppm	17.18	200	[47]
Ag-ZnSnO_3_	HCHO, 50 ppm	26.7	100	[48]
Ag–ZnO/GO	C_2_H_2_, 100 ppm	2.87	250	[49]
Ag/SnO_2_	Ethanol, 50 ppm	135	180	[50]
Ag-TiO_2_	Acetone, 100 ppm	13.9	275	[51]
Ag-WO_3_	NH_3_, 5 ppm	3.29	200	[52]
Ag-ZnSnO_3_	Isopropanol, 1000 ppm	6	150	This work

## Data Availability

Data is contained within the article and Appendix A.

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
