# Peer review of "VOC Gas Sensors Based on Zinc Stannate Nanoparticles Decorated with Silver"

_nanomaterials, 2024, doi:10.3390/nano14241993_

Round 1
Reviewer 1 Report
Comments and Suggestions for Authors
The manuscript reports the synthesis, characterization, and gas-sensing
performance of zinc stannate nanoparticles with and without modification by silver
nanoparticles. The authors showed improved performance of zinc stannate
nanoparticles when modified by silver nanoparticles. The findings have significant
implications for the advancement of VOC sensors. However, the manuscript needs
to be significantly improved.
1. What is the percentage of silver nanoparticles in Zinc stannate? The authors
should provide an EDX analysis to confirm this.
2. Why is it necessary to anneal the spin-coated sample at the high temperature
of 500C? Does annealing at 500C for 15 minutes further change the
crystallinity or morphology of the nanoparticles? The authors should provide
additional XRD characterization results to show if there is any change in the
crystallinity.
3. Is the sensor response stable while passing dry air for a prolonged period?
The authors should provide the data to show the baseline stability.
4. How do the same samples annealed at different temperatures show selectivity
to different VOCs? The authors should explain a probable mechanism based
on their work or similar works reported in the literature.
5. Why does ZTO_500, initially selective and responsive to acetone, show a
decreased response (for acetone) when modified with silver nanoparticles?
How does the modification with silver nanoparticles make it responsive to
Isopropyl alcohol and ethanol?
Author Response
We would like to thank the reviewers for their careful and detailed reports. We appreciate their comments and suggestions and have made every effort to address them in the revised manuscript. The changes made in the revised version are clearly indicated.
- What is the percentage of silver nanoparticles in Zinc stannate? The authors should provide an EDX analysis to confirm this.
Thank you for your comment. Initially, when modifying zinc stannate powder with silver nanoparticles, the percentage of silver was 5 wt.% (50 mg of Ag added to 1 g of ZTO-500). EDX map of Ag distribution and EDS spectrum of ZTO-500_Ag were added to the results (Figure 7).
We added the following text to our revised version (page 6). The EDS spectrum (Figure 7, a) confirms the presence of Ag in the modified sample. At the same time, the percentages of elements determined by the spectrum are 51.5 wt.% (Sn), 24.2 wt.% (Zn), 15.4 wt.% (O), and 8.9 wt.% (Ag). EDX scan (Figure 7, b) shows the uniform distribution of silver particles over the surface of the sensor layer.
- Why is it necessary to anneal the spin-coated sample at the high temperature of 500C? Does annealing at 500C for 15 minutes further change the crystallinity or morphology of the nanoparticles? The authors should provide additional XRD characterization results to show if there is any change in the crystallinity.
Thank you very much for your comment. The methodology of the experiment in the previous version of the manuscript was given in general. In fact, the spin-coated sample was annealed at the same temperatures as the initial powder (300 °C for ZTO-300, 500 °C for ZTO-500 and 700 °C for ZTO-700). Annealing was performed to stabilize the resistance of the sensor sample.
This paragraph has been revised to clarify the annealing process (page 3).
The coated substrates were then annealed at 300 °C (ZTO-300), 500 °C (ZTO-500 and (ZTO-500_Ag) and 700 °C (ZTO-700) for 15 minutes in an air atmosphere.
- Is the sensor response stable while passing dry air for a prolonged period? The authors should provide the data to show the baseline stability.
Thank you for your comment. Based on your comment, we added the following text to our revised version.
The stability of the baseline resistance was studied at a temperature of 250 °C for 12 hours. The results are presented in Figure S2. During this time, the maximum deviation of the resistance from its initial value was 4%, as shown in the figure.
- How do the same samples annealed at different temperatures show selectivity to different VOCs? The authors should explain a probable mechanism based on their work or similar works reported in the literature.
Thank you for your comment. Based on your comment, we added the following text to our revised version (page 10).
A possible explanation for the observed dependencies may be related to the different mechanisms of interaction between metal oxides, acetone, and alcohols. According to XRD results, ZTO-300 and ZTO-500 samples have a similar crystal structure. However, there may be differences in the ratio of zinc, tin, and oxygen ions on the surfaces of these samples. It is known that acetone molecules can be adsorbed on metal cations [37, 38], while alcohol adsorption can occur on both metal cation sites and hydroxyl groups [39, 40]. The high concentration of metal cation sites on ZTO-500 surface, which serve as adsorption sites for acetone molecules, could explain why the response to acetone is higher than the response to alcohol. Conversely, the higher response of ZTO-300 sample to isopropyl alcohol may be due to a higher content of oxygen vacancies on its surface, which can accept adsorbed hydroxyl groups from ambient air. The lower responses of the ZTO-700 sample are associated with the formation of two phases, Zn2SnO4 and SnO2.
- Why does ZTO_500, initially selective and responsive to acetone, show a decreased response (for acetone) when modified with silver nanoparticles? How does the modification with silver nanoparticles make it responsive to Isopropyl alcohol and ethanol?
Thank you for your comment. Based on your comment, we added the following text to our revised version (page 13).
The observed decrease in the response of ZTO-500 to acetone after modification with silver nanoparticles can be caused by the influence of a chemical sensitization. As a result of this process, the concentration of ionosorbed oxygen ions, occupying free metal cation sites on the surface, increases. Therefore, the content of adsorption sites for acetone molecules decreases, therefore, the response to acetone also decreases. At the same time, as discussed earlier, the adsorption of alcohol molecules can occur with the participation of surface hydroxyl groups, so the concentration of adsorbed alcohol molecules remains high. An increase in the concentration of ionosorbed oxygen leads to an increase in the number of electrons formed as a result of the reaction (10) and an increase in the responses to isopropyl alcohol and ethanol.
Reviewer 2 Report
Comments and Suggestions for Authors
In this work, the authors developed a VOC gas sensor based on zinc stannate nanoparticles decorated with silver, where different annealing temperatures were applied to investigate the phase composition of the sample and their sensing properties. The results revealed that the ZTO-300 had a better response to isopropyl alcohol, while ZTO-500 exhibited a better response to acetone. Moreover, Ag decoration contributed to an increase of response. After consideration, I suggest a major revision before being accepted for possible publications.
1. The characterization for Ag decorated Samples should be provided to confirm successful decoration of Ag nanoparticles on the surface of zinc stannate nanoparticles, such as HRTEM and XPS.
2. The device structure should be provided.
3. The gas sensing performance under different gas concentrations should be provided.
4. The stability of gas sensing performance in air and the influence of relative humidity should be investigated.
5. Ag decoration has been widely used in improving gas sensing performance. A performance comparison between this work and other VOC sensors decorated with Ag should be provided to demonstrate the advantages of this work.
Comments on the Quality of English LanguageEnglish should be improved.
Author Response
We would like to thank the reviewers for their careful and detailed reports. We appreciate their comments and suggestions and have made every effort to address them in the revised manuscript. The changes made in the revised version are clearly indicated.
- The characterization for Ag decorated Samples should be provided to confirm successful decoration of Ag nanoparticles on the surface of zinc stannate nanoparticles, such as HRTEM and XPS.
Thank you for your comment. Successful decoration of Ag nanoparticles on the surface of zinc stannate has been confirmed by EDS, EDX, and XPS analysis. The EDS analysis (Figure 7, a) revealed that the concentration of silver nanoparticles is 8.9 wt. %. EDX analysis (Figure 7, b) has confirmed the uniform distribution of silver during the modification of zinc stannate. Study of the Ag3d X-ray photoelectron spectrum (Figure 10) has shown the presence of peaks with binding energies of 374.4 eV (Ag3d3/2) and 368.4 eV (Ag 3d5/2), corresponding to metallic silver.
The figures and explanations are given on pages 7-8.
- The device structure should be provided.
Thank you for your comment. The structure of the device is added to the supporting information in Figure S1. The distance between the electrodes and their width was 200 μm.
- The gas sensing performance under different gas concentrations should be provided.
Thank you for your comment. The dependence of the response of ZTO-500_Ag to isopropyl alcohol at 250°C on the concentration of the target gas (Figure 13) is added in the revised version of the manuscript (page 11).
- The stability of gas sensing performance in air and the influence of relative humidity should be investigated.
Thank you for your comment. Based on your comment, we added following text into our revised version (page 10).
The effect of water vapor on the response of ZTO-500_Ag when detecting isopropyl alcohol at a concentration of 1000 ppm showed that at a relative humidity of 10%, the response was reduced by 17%.
- Ag decoration has been widely used in improving gas sensing performance. A performance comparison between this work and other VOC sensors decorated with Ag should be provided to demonstrate the advantages of this work.
Thank you for your comment. Based on your comment, we added following text into our revised version (page 13).
A comparison of the gas sensing characteristics achieved in this study with the results of other researchers on the development of silver-modified sensors based on metal oxide materials is presented in Table 3. The table shows that silver modification is an effective approach for developing highly efficient sensors. The findings from this study demonstrate the ability to achieve response and recovery at temperatures significantly lower than those reported in most previous studies.
Round 2
Reviewer 2 Report
Comments and Suggestions for Authors
The authors have addressed all my concerns. I recommend its publication at present form.